# Chest Imaging in Systemic Endemic Mycoses

**DOI:** 10.3390/jof8111132

**Published:** 2022-10-27

**Authors:** Célia Sousa, Edson Marchiori, Ali Youssef, Tan-Lucien Mohammed, Pratik Patel, Klaus Irion, Romulo Pasini, Alexandre Mançano, Arthur Souza, Alessandro C. Pasqualotto, Bruno Hochhegger

**Affiliations:** 1Radiology Department, Centro Hospitalar Universitário de São João, 4200-319 Porto, Portugal; 2Radiology Department, Universidade Federal de Rio de Janeiro, Rio de Janeiro 21941-901, Brazil; 3Radiology Department, University of Florida Health Shands Hospital, Gainesville, FL 32608, USA; 4Radiology Department, Faculdade de Medicina de São José do Rio Preto, São José do Rio Preto 15090-000, Brazil; 5Radiology Department, Santa Casa de Misericórdia de Porto Alegre, Porto Alegre 90020-090, Brazil

**Keywords:** endemic mycoses, histoplasmosis, coccidioidomycosis, *cryptococcosis*, blastomycosis, paracoccidioidomycosis, computed tomography

## Abstract

Endemic fungal infections are responsible for high rates of morbidity and mortality in certain regions of the world. The diagnosis and management remain a challenge, and the reason could be explained by the lack of disease awareness, variability of symptoms, and insidious and often overlooked clinical presentation. Imaging findings are nonspecific and frequently misinterpreted as other more common infectious or malignant diseases. Patient demographics and clinical and travel history are important clues that may lead to a proper diagnosis. The purpose of this paper is to review the presentation and differential diagnosis of endemic mycoses based on the most common chest imaging findings.

## 1. Introduction

Fungi may have a widespread or endemic geographic distribution. Immunocompromised patients are most affected by ubiquitous fungi, such as *Aspergillus*, *Candida*, *Mucor*, *Cryptococcoci neoformans*, and *Pneumocystis jiroveci*. Immunocompetent individuals or those with chronic conditions may be susceptible to endemic mycoses. Overall, patients with reduced immunity are more vulnerable to complications and disseminated disease from endemic and ubiquitous fungal infections. Additionally, endemic mycoses can be divided into two groups, such as implantation or subcutaneous mycoses in which the pathogen generally infects transcutaneous wounds and systemic mycoses in which the respiratory tract is the primary route of transmission into the human host, i.e., aerogenic and dust inhalation (construction, farming, landscaping, excavation) [1,2]. Systemic endemic mycoses are a group of dimorphic fungi prevalent in specific geographical locations. *Histoplasma capsulatum*, *Coccidioides* spp., *Blastomyces dermatitidis*, *Cryptococcosis gattii, and Paracoccidioides brasiliensis* are the primary pulmonary fungal pathogens of otherwise healthy people. *Histoplasma capsulatum* is found worldwide, but particularly in North, Central, and South America. Pulmonary coccidioidomycosis or valley fever is caused by the dimorphic fungi *C. immitis* and *C. posadasii*. It is endemic in the southwestern parts of the USA (California, Arizona, Utah, New Mexico, and Nevada) and parts of Central and South America (Brazil, Argentina, Mexico). Blastomycosis is mainly reported in North America and Africa. Cryptococcosis (*C. gattii*) is endemic in the Pacific Northwest, Central and South America, Asia, and India. Paracoccidioidomycosis (*P. brasiliensis*) is endemic in Latin America, particularly Brazil, Colombia, Venezuela, and Argentina [3,4,5]. Additionally, talaromycosis and emergomycosis are two other important endemic systemic mycoses; however, they are more frequently reported as opportunistic infections in immunocompromised patients, particularly in advanced HIV disease. Talaromycosis, caused by *Talaromyces marneffei*, is endemic to north-eastern India, southeast Asia, and southern China. Emergomycosis are endemic to Africa (*Emergomyces pasteurianus*, *Es africanus*), Europe (*Es pasteurianus*, *Es europaeus*), North America (*Es canadensis*), and Asia (*Es pasteurianus*, *Es orientalis*) [6,7].

The majority of infected people remain asymptomatic or develop self-limiting respiratory symptoms (up to 6 weeks). The clinical presentation of these granulomatous diseases can vary from asymptomatic to disseminated infection and depends on the amount of environmental exposure, virulent strain, host’s immune status, and extremes of age. Symptoms are often subacute but may have an acute presentation. Cough, fever, chills, anorexia, weight loss, fatigue, headache, and chest pain are all possible clinical symptoms. Patients may also manifest dermatological symptoms, such as erythema nodosum or multiforme, and rheumatological manifestations. The severe disease form may spread hematogenously to bones, skin, joints, and central nervous system [1,4].

The diagnosis of systemic endemic mycosis is challenging and frequently misinterpreted for other diseases (e.g., bacterial or viral pneumonia, tuberculosis, sarcoidosis, lung cancer, metastases). Both clinicians and radiologists may be unfamiliar with the disease manifestations, especially those from non-endemic areas. Familiarization with these diseases by health professionals is of crucial importance as they are becoming increasingly relevant worldwide due to traveling and immigration [4,8].

Imaging findings are nonspecific and overlap with both other non-fungal diseases and among the numerous fungal pathogens. The aim of this paper is to review the most common imaging presentation and differential diagnoses of systemic endemic mycoses.

## 2. Imaging Findings

### 2.1. Lung Nodule or Mass

Acute or chronic fungal infections may manifest as solitary or multiple lung nodules or masses. The most frequent pathogens are histoplasmosis, coccidioidomycosis, *cryptococcosis*, and blastomycosis [4]. The nodules have a nonspecific appearance, may be ill or well-defined, have regular or spiculated borders, and may also demonstrate cavitation or ground glass halo ***(***Figure 1). Additionally, a presenting dominant nodule or mass may be associated with other satellite nodules or bronchovascular beading [9]. Three-in-bud opacities are less common but may be seen in cases of bronchiolitis. More severe infections may present with multiple solid or cavitary nodules with a tendency toward confluence (Figure 2) [4,10]. In histoplasmosis, cryptococcosis, and coccidioidomycosis, the size of the nodules frequently ranges from less than 10 to 30 mm and they have a lower lobe predominance. The nodules in histoplasmosis and cryptococcosis are also predominantly peripherally located (Figure 3) [4,5,10,11,12,13]. Lung masses are the second most common finding in blastomycosis (after lobar and segmental consolidation), and they may measure up to 10 cm in diameter [4,14]. Pulmonary granulomata resulting from histoplasmosis may continue to enlarge over time (average of 1.7 mm per year) due to proliferation of fibrosis at the periphery of the nodule, secondary to an abnormal host response [9]. 

Hilar and mediastinal lymphadenopathies are typically visualized in histoplasmosis and coccidioidomycosis, which may be bulky, especially in the latter [4]. In acute infection, Fluorine 18 (18F)–fluorodeoxyglucose (FDG) PET/CT shows avid uptake in both the pulmonary infection and lymph nodes but with a greater degree of pulmonary uptake. During the subacute phase of infection, the pulmonary nodules or masses demonstrate a decrease in FDG activity more quickly than the adenopathy (Figure 4 and Figure 5). This is the opposite of lung cancer in which the malignancy usually retains higher FDG avidity. This finding, also known as the flip-flop fungus sign, discloses the benign granulomatous disease nature, particularly associated with histoplasmosis [15]. As the infection heals, the infected mediastinal lymph nodes and pulmonary granulomata can calcify, showing central (target) or diffuse calcification, a typical finding in histoplasmosis [4,8,9]. Hilar and mediastinal lymphadenopathy and pulmonary calcified nodules are not typical findings in coccidioidomycosis, cryptococcosis and blastomycosis [12,13,14]. 

Although nodules or masses can predominate on imaging findings, mixed patterns and additional findings are frequently visualized: consolidations, ground-glass opacities, bronchovascular and septal thickening, and airway disease. Pleural effusions are not a common finding but may be present. More rarely, other pulmonary complications, such as pneumothorax, bronchopleural fistula, and lung abscess, can be found [4,5,8,11].

The main concern in differential diagnosis is the exclusion of malignancy (primary lung cancer and metastases). Other differentials include tuberculosis, nontuberculous mycobacteria, sarcoidosis, vasculitis, and further infectious fungi, such as semi-invasive or invasive aspergillosis, mucormycosis, and candidiasis, especially in immunocompromised patients [4,8,11,16].

### 2.2. Non-Resolving Pneumonia

Acute fungal infections can present clinically with flu-like symptoms or as community acquired bacterial pneumonia with fever, cough, pleuritic chest pain, myalgia, and headache. Lobar, segmental or patchy multifocal consolidations involving several lobes concurrent with hilar and mediastinal lymphadenopathy are common findings of coccidioidomycosis and histoplasmosis (Figure 6). A tree-in-bud pattern may also be present. Progressive cavitation of the consolidations can occur as the disease progresses. The consolidations in coccidioidomycosis tend to be unilateral, basilar, or perihilar [11,17,18]. Additionally, nodules or masses may evolve at sites of prior consolidation, and eventually can form residual granuloma [4,9]. 

The most prevalent pattern of blastomycosis is patchy, ill-defined consolidations with air bronchograms. The disease may be unilateral or bilateral, and areas of confluence can become quite extensive. These findings may also be visualized alongside the presence of large nodules or masses [19,20].

Segmental or lobar consolidations are also recognized features of cryptococcosis, usually coexisting with parenchymal nodules and masses (Figure 7) [21,22].

Associated hilar and mediastinal lymphadenopathies are frequent in histoplasmosis and coccidioidomycosis. Pleural or pericardial effusions are not a common finding but may be present. When this complication is present, it occurs most frequently secondary to a hypersensitivity reaction to the fungus antigens. It develops mainly in younger patients and is self-limited in the vast majority of cases. Less frequently, pleural or pericardial effusions can also occur during immune reconstitution syndrome, due to an excessive inflammatory response against the pathogen after a rapidly restored immune system. This finding has been especially reported in histoplasmosis infection in immunocompromised patients (HIV, immunomodulatory therapy) [23,24]. Additionally, empyema may complicate some cases of pleural effusions [4].

The differential diagnoses include community-acquired pneumonia, malignancy, tuberculosis, aspiration, airway invasive aspergillosis, and organizing pneumonia [8].

### 2.3. Chronic Cavitating Disease

Chronic infections, mainly associated with histoplasmosis and coccidioidomycosis, may develop over time into scarring or cavitating disease. The findings are similar to tuberculosis and most frequently appear in the upper lobes, especially in the apical and posterior segments (Figure 8). Imaging depicts chronic consolidation with progressive cavitation. The cavity may present wall thickening, air-fluid level, internal fungus balls, and may enlarge or collapse, ultimately resulting in volume loss [9,11]. Pleural thickening adjacent to cavitary lesions is also common. Additionally, coccidioidomycosis may evolve thin-walled cavities, a finding termed grape-skin cavities [25]. Calcified lymph nodes may be present in histoplasmosis, but lymphadenopathy is usually absent. Conversely, lymphadenopathy is common in coccidioidomycosis, especially in the acute phase. Other pulmonary findings are recurrently present, such as patchy consolidations, ground-glass opacities, nodules, bronchiectasis, fibrosis, and architectural disorganization [4].

Complications are uncommon and include pneumothorax, fistula formation, broncholiths, empyema, hemoptysis, atelectasis, and pleural and chest wall invasion [4,8]. Involvement of the airways (trachea and bronchi) or larynx can also occur, especially in chronic coccidioidomycosis and paracoccidioidomycosis [11,26].

Chronic pulmonary aspergillosis (CPA) is an important differential diagnosis and usually occurs in non-immunocompromised patients with prior or current lung disease. Chronic cavitating PA is the most common form of CPA, and if untreated it may progress to chronic fibrosing PA (CFPA). Subacute invasive aspergillosis is another form of CPA, occurring in mildly immunocompromised patients (such as those with a history of alcoholism, diabetes mellitus, connective tissue disorders) and usually has a more rapid progression. Imaging findings include necrotic nodules or masses, cavities with thin or thick walls, perhaps containing intraluminal debris and aspergilloma. Pericavitary infiltrates, consolidation, and fibrosis may also be present (Figure 9). Severe fibrotic destruction involving at least two lobes is a hallmark of CFPA [27].

Additional differential diagnoses include postprimary pattern of tuberculosis, lung cancer, nontuberculous mycobacterial infection, Actinomyces, mucormycosis, and Nocardia [8].

### 2.4. Disseminated Infection

Endemic mycosis is a self-limited disease in the vast majority of immunocompetent individuals. Immunosuppressed patients are at risk for disseminated disease: HIV, transplant recipients, biologic modifying agents, hematological malignancies, corticosteroid therapy, and extremes of age. Symptoms may include fever, malaise, weight loss, septicemia, severe respiratory distress, coagulopathy, and multiorgan failure. This form of disease presentation is most frequently associated with histoplasmosis and coccidioidomycosis. Chest imaging may depict diffuse air-space opacities, multiple consolidations, acute respiratory distress syndrome (ARDS), and diffuse miliary micronodules (Figure 10). The severe chest imaging findings represent the spread of infection and the difficulty of controlling fungal proliferation in immunosuppressed patients. Extrathoracic dissemination may affect any organ, including skin, lymph nodes, heart, adrenal glands, gastrointestinal tract, bones, joints, and central nervous system [5,9,28,29].

Talaromycosis and emergomycosis are endemic mycoses that occur predominantly in immunosuppressed patients and frequently manifest as multiorgan disseminated diseases. Talaromycosis can involve the upper and lower respiratory tract. Although rare, the infection of the pharynx, larynx, trachea, and bronchi is a distinctive manifestation, and frequently accompanied by cervical lymphadenopathy. Chest imaging findings are diverse and may include patchy consolidations, ground-glass changes, multiple nodules, cavitary disease, pleural effusions, and lymphadenopathy [6]. Patients with emergomycosis commonly manifest widespread polymorphic skin lesions at clinical presentation (umbilicated papules, verrucous lesions, nodules, erythema, and hyperkeratotic plaques). Chest imaging findings are also highly nonspecific and can include diffuse and focal infiltrates, consolidation, lobar atelectasis, pleural effusions, and hilar lymphadenopathy [7]. 

Differential diagnoses include miliary tuberculosis, hematogenous metastases, and pneumocystis pneumonia (Figure 11).

### 2.5. Mixed Pattern

Paracoccidioidomycosis is endemic in Latin America, with the greatest number of cases originating in Brazil. However, several cases have been reported in Europe and North America, particularly among travelers and immigrants. The initial infection is similar to the primary complex of tuberculosis and may have a self-limited path controlled by the host immune response or may progress to symptomatic disease. The two leading clinical forms of paracoccidioidomycosis are the acute form or juvenile type and the chronic form or adult type. The adult form may manifest clinically several years after the inhalation of the infectious particles and accounts for the vast majority of the disease presentation (90% of cases). The disease installation is typically insidious and classically characterized by pulmonary involvement in more than 90% of patients and chronic development of mucocutaneous lesions in approximately 50% of patients. The dissociation between inoculation and proliferation of the fungus and the initial clinical symptoms is responsible for the frequent extensive imagiological findings and severe lung damage in the first exams. Pulmonary fibrosis may be present from the time of diagnosis in 32% of patients, even with indirect signs of pulmonary hypertension and *cor pulmonale* [30,31,32]. Imaging findings are nonspecific and may include: patchy ground-glass opacities, consolidation, nodules of variable sizes, interlobular septal thickening, intralobular lines, cavitation, and manifestations of fibrosis, such as traction bronchiectasis, paracicatricial emphysema, architectural distortion (Figure 12 and Figure 13). A miliary pattern and lung opacities with “halo” and “reversed halo” signs are also described. These changes are often combined and tend to be bilateral, symmetrical, and involve at least one-third of the lung parenchyma. The disease regularly affects all lung zones; however, a tendency toward the middle zones configuring a butterfly wing pattern may also suggest the diagnosis. Residual fibrotic changes may persist in 60% of patients. Enlarged mediastinal or hilar lymph nodes and pleural effusions are uncommon in the chronic form of the disease. Patients may also manifest extrapulmonary findings, such as tracheal involvement, pneumothorax, joint and osseous lesions. Tracheal infection may manifest as irregular circumferential wall thickening with submucosal nodules, and it may result from direct contact with infected sputum, lymphatic drainage, or hematogenous dissemination. Paracoccidioidomycosis is uncommon in immunocompromised patients, such as those with hematologic malignancies, HIV infection, or transplant recipients. Differential diagnoses include community-acquired pneumonia, other fungal diseases, tuberculosis, and malignancy [5,33,34]. 

### 2.6. Central Bronchiectasis and Asthma

Allergic bronchopulmonary aspergillosis (ABPA) is an uncommon hypersensitivity reaction to fungal infection that affects immunocompetent individuals. It has a ubiquitous distribution and does not belong to the family of endemic mycosis; however, this disease is worth mentioning. ABPA is classically portrayed in asthmatic patients but also occurs in individuals with cystic fibrosis, lung transplantation, or Kartagener’s syndrome. Asthma-induced mucosal damage of the proximal airways enables the proliferation of the fungus, creating an additional mucosal injury, mucus production, and bronchiectasis [8]. Imaging findings can be either transient or permanent and include predominant central bronchiectasis, mucous plugging, consolidation or non-homogeneous infiltrations frequently surrounding the secretion-filled bronchiectasis, centrilobular nodules, parenchymal scarring, fibrosis, mosaic perfusion, and air trapping on expiration. Plain films show tubular and branching opacities with peri-hilar predominance, related to bronchiectasis with mucoid impaction (finger in glove sign). High-attenuation mucus plugging on CT is a pathognomonic feature of ABPA, noticed in 28% of patients, and represents metallic ions and calcium produced by the fungus (Figure 14). Lobar or segmental collapse is not uncommon [8,35].

### 2.7. Additional Intra and Extrathoracic Findings

Fibrosing mediastinitis is a rare, delayed and potentially life-threatening fibroinflammatory process associated most commonly with *H capsulatum* infection (usually years earlier). Most cases are seen in young patients and are thought to be caused by an abnormal reaction to fungus antigens in genetically susceptible individuals. On CT, it appears as an infiltrative mediastinal fibrous mass, replacing the fat and encasing the adjacent airways and vascular structures, usually with calcified lymph nodes. This complication commonly causes compression of the superior vena cava (with resulting vena cava syndrome), pulmonary arteries and veins, esophagus, and bronchi. The main causes of morbidity and mortality are related to pulmonary arterial hypertension and *cor pulmonale* [36].

Signs of previous histoplasmosis infection: calcified pulmonary nodules (dense or target calcification), broncholithiasis, hepatic and splenic granulomata [4].

Involvement of the larynx and airways: laryngeal, tracheal, and bronchial disease is a chronic disease presentation, especially in coccidioidomycosis and paracoccidioidomycosis, usually associated with parenchymal disease and with disseminated disease, and most commonly results from direct invasion. Additionally, upper and lower respiratory tract infection is also a distinctive manifestation of talaromycosis. Imaging findings include wall thickening, endoluminal nodules, stenosis, and obstruction. Lymph nodes may also compress and erode into the airways, causing obstruction. Complications of airway obstruction include postobstructive atelectasis or pneumonia, fistula formation, hemoptysis, and lithoptysis (expectoration of broncholiths) [6,33,37].

## 3. Conclusions

We described the most frequent chest imaging findings of endemic mycoses, such as the presence of lung nodules or masses, non-resolving pneumonia, chronic cavitating disease, and disseminated infection, with discussion of the differential diagnoses (Table 1). However, a mixed pattern or a combination of multiple findings are frequently observed and may be focal, multifocal, or diffuse. Immunocompromised patients often present with mixed, diffuse, and severe chest findings. As the geographic distribution of mycoses is spreading, familiarity with the imaging findings, along with a proper clinical, occupational/recreational, and travel history are essential to make an appropriate diagnosis.

## Figures and Tables

**Figure 1 jof-08-01132-f001:**
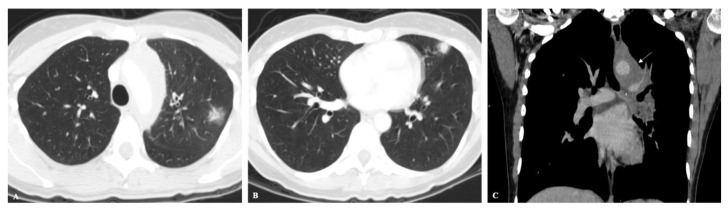
(**A**,**B**) Axial CT images showing peripherally located solid nodules with ground-glass halo in the upper left lobe, in a patient with acute histoplasmosis infection. (**C**) Coronal CT image depicts left mediastinal and hilar enlarged lymph nodes (arrow).

**Figure 2 jof-08-01132-f002:**
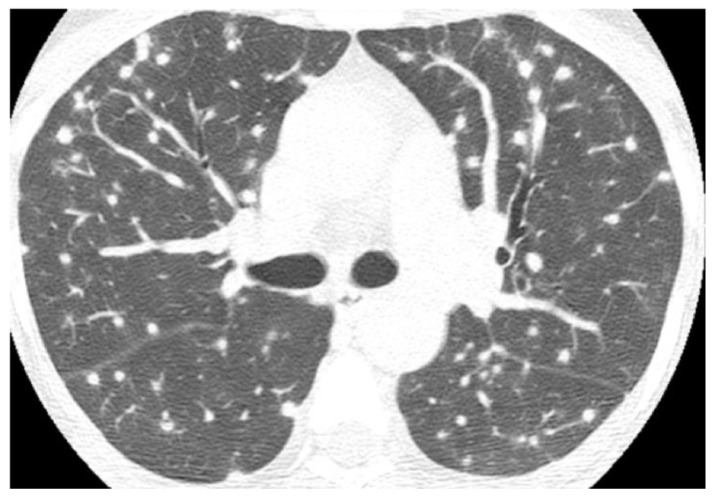
Axial CT image in a patient with histoplasmosis shows numerous bilateral and randomly distributed solid nodules, some of them with peripheral ground-glass halos.

**Figure 3 jof-08-01132-f003:**
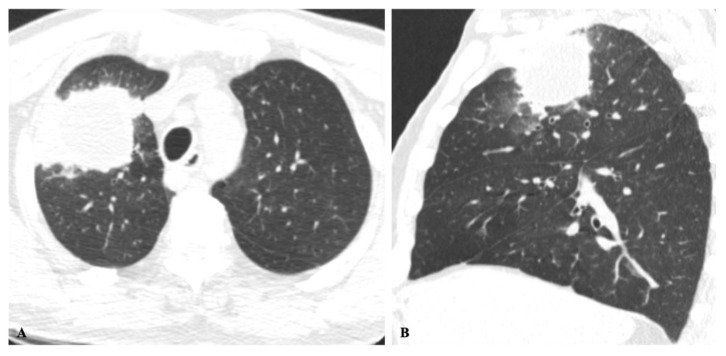
Cryptococcosis in a 63-year-old man with clinical history of colon cancer, heavy smoking and cough for 3 months. (**A**) Axial and sagittal (**B**) images show an irregular solid mass in the right upper lobe adjacent to the pleura and ground-glass halo.

**Figure 4 jof-08-01132-f004:**
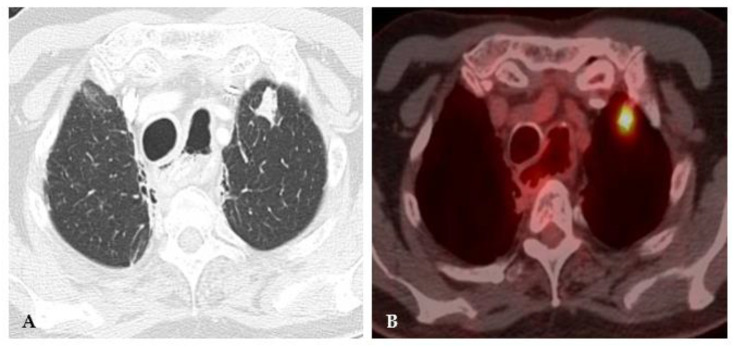
Acute histoplasmosis infection. (**A**) Axial CT image depicts a left upper lobe nodule. (**B**) FDG PET/CT shows avid uptake in the pulmonary infection.

**Figure 5 jof-08-01132-f005:**
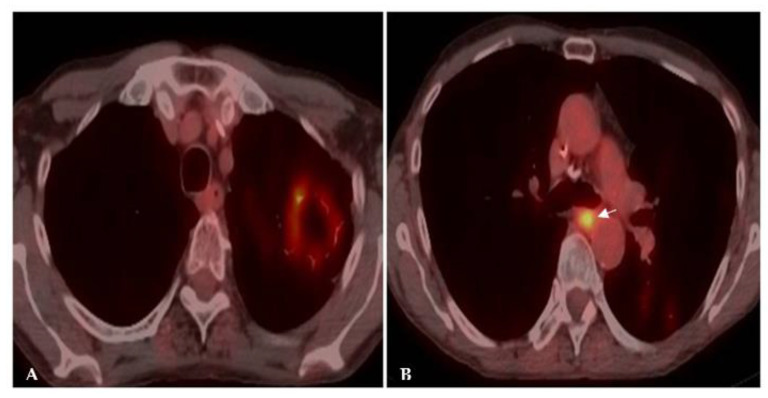
Chronic histoplasmosis infection. FDG PET/CT images show higher FDG activity in a mediastinal draining lymph node (arrow) in (**B**) than in the left upper lobe cavitary mass (**A**). The flip-flop fungus sign discloses the benign granulomatous nature of the disease.

**Figure 6 jof-08-01132-f006:**
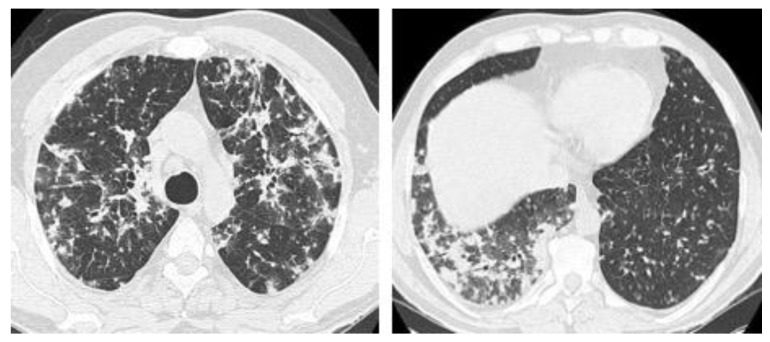
Acute histoplasmosis infection. Axial CT images from upper (**left image**) and lower levels (**right image**) showing bilateral peribronchovascular consolidations and groundglass opacities, and ill-defined centriacinar nodules.

**Figure 7 jof-08-01132-f007:**
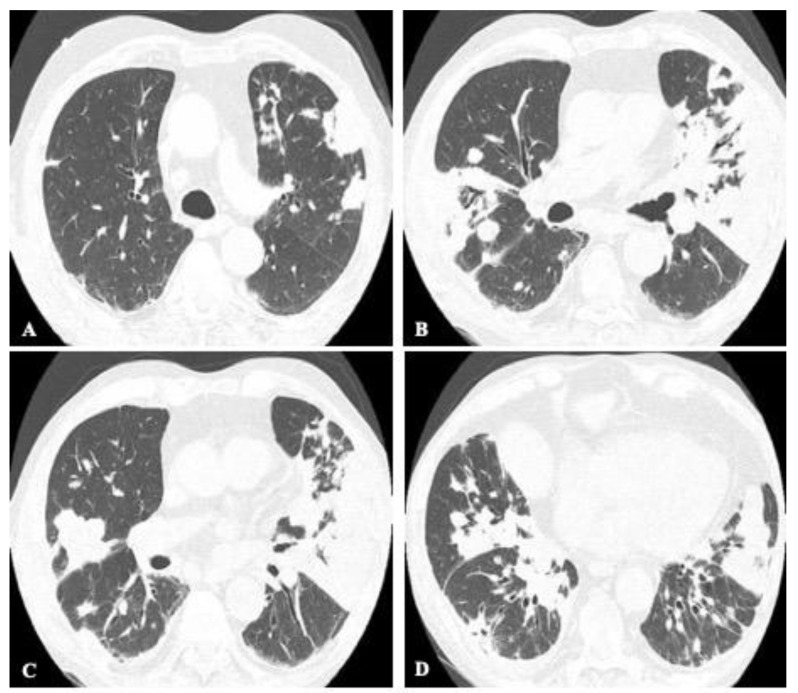
72-year-old male with acute cryptococcosis. Axial CT images from upper to lower levels (**A**–**D**) displaying bilateral consolidations with air bronchograms and multiple nodules.

**Figure 8 jof-08-01132-f008:**
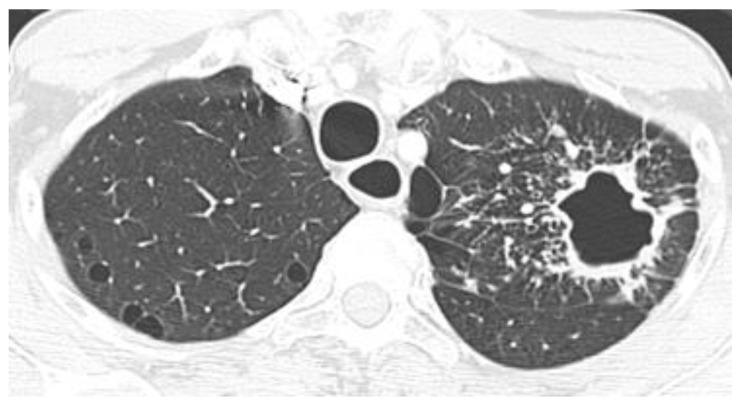
Chronic histoplasmosis infection. CT axial image depicts a spiculated cavity with thick walls in the left upper lobe.

**Figure 9 jof-08-01132-f009:**
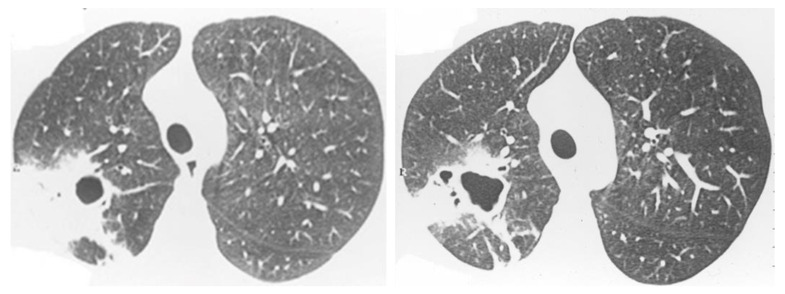
Subacute invasive aspergillosis in a 35-year-old male with HIV infection. CT axial images from an upper and lower levels (left to right) show an irregular cavitary mass in the right upper and lower lobes transgressing the major fissure.

**Figure 10 jof-08-01132-f010:**
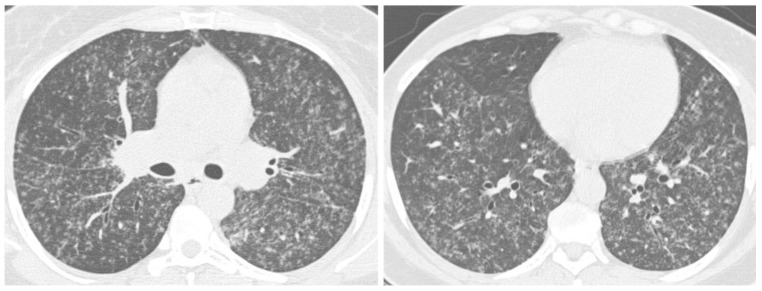
Diffuse miliary nodules in an immunosuppressed patient with histoplasmosis. Axial CT images from upper and lower levels (from **left** to **right**) depict diffuse micronodules randomly distributed in both lungs.

**Figure 11 jof-08-01132-f011:**
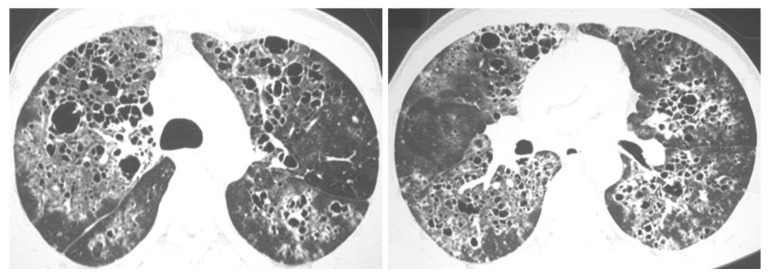
Pneumocystis pneumonia in a patient with acquired immunodeficiency syndrome (AIDS). Axial CT images from upper and lower levels (from **left** to **right**) reveal diffuse bilateral ground-glass opacities with some peripheral sparing. Multiple bilateral pneumatoceles of varying size, shape and wall thickness are also visualized.

**Figure 12 jof-08-01132-f012:**
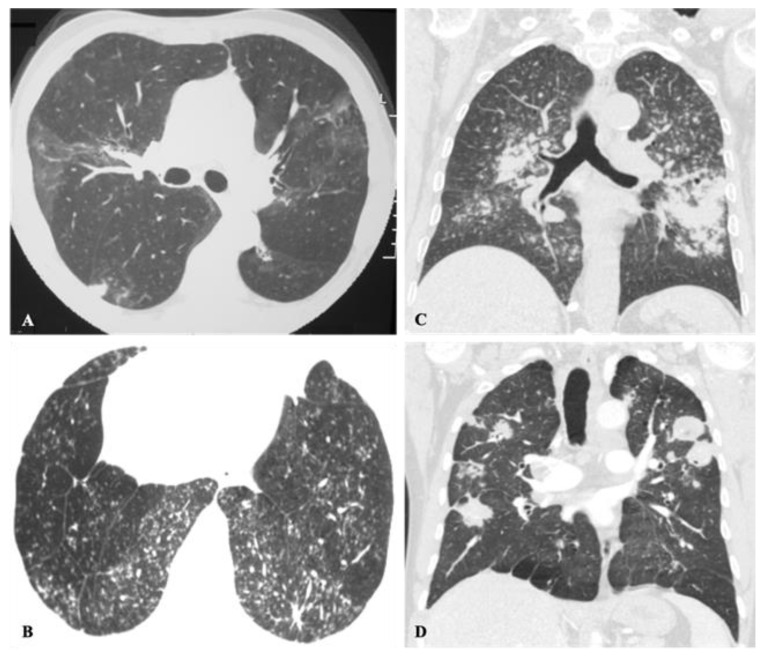
Imaging findings in different patients with paracoccidioidomycosis. The changes are often bilateral, affecting both central and peripheral lung areas and may predominate in the middle zones. The changes are often combined; however, some patterns may prevail. Imaging findings may include patchy bilateral predominant ground-glass opacities (**A**), multiple bilateral small nodules (**B**), bilateral consolidations with associated nodules (**C**), and a dominant pattern of large bilateral nodules (**D**).

**Figure 13 jof-08-01132-f013:**
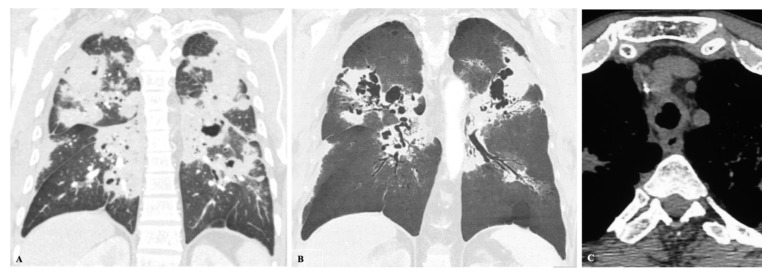
52-year-old man with paracoccidioidomycosis, presenting with a history of 1 year of shortened of breath that had worsened in the last 20 days. (**A**) coronal CT and (**B**) coronal minimum-intensity projection images depict bilateral and symmetrical consolidations and cavitations in a “butterfly wing” pattern, nodules, and ground-glass opacities. CT also reveals tracheal infection, with irregular circumferential thickening of the wall (**C**). Diagnosis was established by transbronchial biopsy.

**Figure 14 jof-08-01132-f014:**
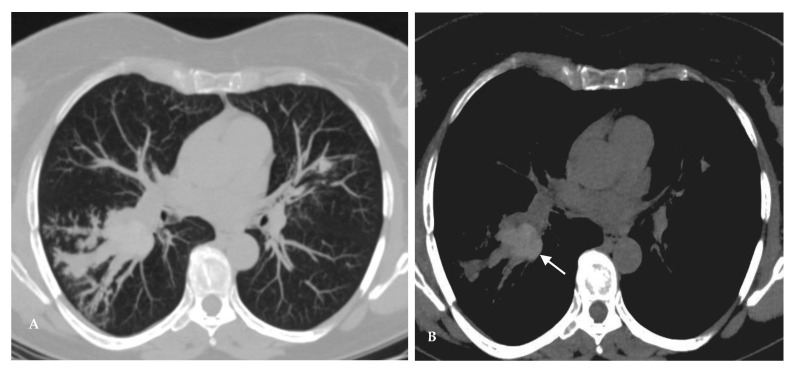
39-year-old female with ABPA and asthma. (**A**) Axial CT image depicts right lower lobe varicose bronchiectasis with mucoid impaction. (**B**) High-attenuation mucus plugging on CT (arrow): a pathognomonic feature of ABPA.

**Table 1 jof-08-01132-t001:** Chest imaging findings of systemic endemic mycoses.

Lung Nodule or Mass		Non-Resolving Pneumonia		Chronic Cavitating Disease		Disseminated Infection		Bronchiectasis & Asthma	
Imaging Clue	Dx	Imaging Clue	Dx	Imaging Clue	Dx	Imaging Clue	Dx	Imaging Clue	Dx
Adenopathy	Coccidioidomycosis Histoplasmosis	Consolidation + large nodules/masses	Blastomycosis Cryptococcosis Paracoccidioidomycosis	Grape-skin cavities + Lymphadenopathy	Coccidioidomycosis	Miliary ARDS Extrathoracic	++ Histoplasmosis Coccidioidomycosis	High-attenuation mucus plugging Finger in glove	ABPA
Lung Mass	Cryptococcosis Blastomycosis	Adenopathy	Coccidioidomycosis Histoplasmosis	Calcified nodes	Histoplasmosis				
Flip-flop node SUVmax > lung mass	Granulomatous Infection								

SUV, standardized uptake value; ARDS, Acute respiratory distress syndrome; ABPA, Allergic bronchopulmonary aspergillosis; ++, most frequent agents.

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
