# Peer review of "Chest Imaging in Systemic Endemic Mycoses"

_jof, 2022, doi:10.3390/jof8111132_

Round 1

Reviewer 1 Report

It is a great revision, important in the current epidemiological context.

I would like to suggest some details that are minor, but for me important.

1.            LINE 42: Paracoccidioidomycosis is not the most important systemic mycosis in Latin America, first by frequency is Histoplsmosis. Tip: change "is the most frequent mycosis"

2.            LINE 135: Progression from consolidation to cavitation can also occur from the onset of the disease in chronic pulmonary histoplasmosis. Tip: Add the concept after  "speciality in disease pulmonary"

3.            LINEA 147: Pleural or pericardial effusion can also occur during immune reconstitution syndrome after partial recovery of the immune system, especially in Histoplasmosis. It also occurs in 17% of patients with Paracoccidioidomycosis.  ( REF: Clin Infect Dis 2020;70(4):643-652.   Med Mycol Case Rep;23:62-64.  Semin Res Crit Care Med 2008;29:182-197 )

4.            LINE 174: Involvement of the airway (trachea, bronchi) or larynx can also occur and frequently in Paracoccidioidomycosis. Add (REF: Mycoses 2012;56:189-199.   Med Mycol 2012:51:313 -318.)

5.            Line 254: Pulmonary fibrosis may be present from the time of diagnosis in 32% of patients, even with indirect signs of pulmonary hypertension and cor pulmonale.  Add (REF    Clin Infect Dis 2003;37:894-904.   Semin Res Crit Care Med 2008;29:182-197.  Current Perspectives from Brazil. Open Microbiol J 2017;11:224-282 )

6.            Line 12. I think this X-ray image is not very classic of paracoccidioidomycosis. I am sure that, with the Brazilian authors of the article, a more classic image with mixed commitment of bases and middle fields can be used. Definitely the image used looks more like tuberculosis.

7.            I suggest adding in TABLE 1: Non-resolving Pneumonia, in Consolidation + large nodules/masses: Paracoccidioidomycosis to Blasto and Crypto.

Author Response

Dear Reviewer,

Thanks for the comments and suggestions. 

The authors made the suggested changes:

- LINE 42: changed accordingly.

-LINE 135: In line 139 there is already a comment regarding the evolution from consolidation to cavitation in chronic disease.

-Line 147: changed accordingly.

-Line 174: changed accordingly.

-Line 254: added.

-Line 12: We added a new figure to demonstrate better the mixed imaging pattern of paracoccidioidomycosis.

We also added the suggestion made in the table.

Best regards

Reviewer 2 Report

Your manuscript focused on chest images in the endemic mycoses is interesting for clinicians. Please note that the endemic mycoses englobe two distinct groups of fungal infections: the implantation and the systemic endemic fungal infections. Please explain this at introduction.

I suggest changing the manuscript title to; Chest imaging in systemic endemic mycoses Because you are not considering the endemic implantation mycoses, such as mycetoma, sporotrichosis, chromoblastomycosis, and others.

Author Response

Dear Reviewer,

Thanks for the comments and suggestions.

We changed the manuscript title as suggested.

We also add a comment in the introduction regarding  "implantation and systemic endemic fungal infections" in line 31.

Our best regards